

# A unified data set of airborne cloud remote sensing using the HALO Microwave Package (HAMP)

Heike Konow[1], Marek Jacob[2], Felix Ament[1,3], Susanne Crewell[2], Florian Ewald[4], Martin Hagen[4], Lutz Hirsch[3], Friedhelm Jansen[3], Mario Mech[2], and Bjorn Stevens[3]

[1]Universität Hamburg, Hamburg, Germany
[2]Institute for Geophysics and Meteorology, University of Cologne, Cologne, Germany
[3]Max Planck Institute for Meteorology, Hamburg, Germany
[4]German Aerospace Center DLR, Oberpfaffenhofen, Germany

*Correspondence to:* Heike Konow (heike.konow@uni-hamburg.de)

**Abstract.** Cloud properties and their environmental conditions were observed during four aircraft campaigns over the North Atlantic on 37 flights. The Halo Microwave Package (HAMP) was deployed on the German research aircraft HALO (High Altitude LOng range research aircraft) during these four campaigns. HAMP comprises microwave radiometers with 26 channels in the frequency range between 20 and 183 GHz and a 35 GHz cloud radar. The four campaigns took place between
5 December 2013 and October 2016 out of Barbados and Iceland. Measured situations cover a wide range of conditions including the dry and wet season over the tropical Atlantic and the cold and warm sectors of mid-latitude cyclones. The data set we present here contains measurements of the radar reflectivity factor and linear depolarization ratio from cloud radar, brightness temperatures from microwave radiometers, and atmospheric profiles from dropsondes. It represents a unique combination of active and passive microwave remote sensing measurements and 525 in-situ measured dropsonde profiles.
The data from these different instruments are quality controlled and unified into one common format for easy combination of data and joint analysis. The data are available from the CERA database for the four campaigns individually (https://doi.org/10.1594/WDCC/HALO_measurements_1, https://doi.org/10.1594/WDCC/HALO_measurements_2, https://doi.org/10.1594/WDCC/HALO_measurements_3, https://doi.org/10.1594/WDCC/HALO_measurements_4). This data set allows for analyses to get insight into cloud properties and atmospheric state in remote regions over the tropical and mid-latitude Atlantic.
In this paper, we describe the four campaigns, the data, and the quality control applied to the data.

## 1 Introduction

Clouds in the planetary boundary layer over oceans have been identified as one of the largest contributors to intermodel spread in climate sensitivity (Sherwood et al., 2014) and more detailed observations are needed for model improvement (Bony et al., 2015). Modern satellites provide detailed insights in cloud structures and their microphysical characteristics with almost full
coverage of the Earth. Microwave frequencies are especially suited for cloud and precipitation remote sensing as they can penetrate the full atmospheric column in contrast to solar and infrared remote sensing which are mainly limited to thin clouds and cloud top regions. However, most microwave satellite products are limited in resolution, coverage and sensitivity. Passive





microwave satellite instruments have footprints of several tens of kilometers (Elsaesser et al., 2017) while active microwave instruments are confined to narrow scan regions, limited vertical resolution and sensitivity, i.e. the Global Precipitation Mission (GPM) 35 GHz radar has a minimum detectable signal of 12 dBZ (Skofronick-Jackson et al., 2013) typical for light precipitation. Therefore, the synergy of airborne active and passive microwave remote sensing has mostly been used for liquid and

solid precipitation in the preparation and validation phase of the Global Precipitation Mission (GPM; e.g. Houze et al., 2017; Skofronick-Jackson et al., 2013). A better sensitivity in respect to non-precipitating clouds is reached with about -27 dBZ by the Cloudsat radar being part of the A-Train (Stephens et al., 2002), however, the vertical pulse resolution of about 500 m becomes problematic for boundary layer clouds.

Ground-based remote sensing stations can provide measurements with high temporal and vertical resolution but are limited

to few locations and these are almost exclusively on land. Airborne remote sensing bridges this gap between coarse satellite observations and stationary ground-based measurements. The HALO aircraft (High Altitude LOng range research aircraft, Krautstrunk and Giez, 2012; Wendisch et al., 2016) with its long range, high ceiling and the possibility for carrying a heavy payload is a great platform to explore clouds in maritime conditions over the ocean with highly resolved measurements. The HALO Microwave Instrument Package (HAMP, Mech et al., 2014) has been developed to allow for in-depth remote sensing of

clouds and their microphysical properties. It consists of a 35 GHz cloud radar and a 26 channel microwave radiometer covering a spectral range from 20 to 183 GHz.

In this article, data sets from four campaigns performed in different climate regimes are presented. The first campaign, NARVAL1 (Next-generation Aircraft Remote sensing for VALidation Studies) consisted of two sub-campaigns: NARVAL-South and NARVAL-North. NARVAL-South in December 2013 focused on observations of shallow convection in the trade

wind region of the eastern tropical Atlantic during dry season and consisted of eight research flights out of Barbados (Fig. 1 (a)). During NARVAL-North in January 2014, post-frontal regimes over the extra tropical North Atlantic were observed with seven research flights. Center of operations was Keflavik, Iceland (Fig. 1 (b)). An overview of all NARVAL1 flights and measurements is provided by Klepp et al. (2014). NARVAL2 succeeded this first demonstrator mission in August 2016. During this campaign, HALO again was deployed to Barbados (Fig. 1 (c)). NARVAL2 measurements complement the NARVAL-South measurements

by observations of the atmosphere over the tropical Atlantic during the wet season (Stevens et al., 2018). Directly following the NARVAL2 mission, the same HALO payload was flown during the NAWDEX (North Atlantic Waveguide and Downstream Impact Experiment) campaign (Schäfler et al., 2018). Here, HALO again was deployed to Keflavik, Iceland, for five weeks in September and October 2016 (Fig. 1 (d)). HAMP observations taken during NAWDEX give insight into convective clouds and their surroundings in the frontal regions and warm sector of mid-latitude cyclones. In this paper a unified data set comprising

HAMP measurement data from the four campaigns described above is presented and made available.

The measurements were rigorously quality controlled to ensure temporal collocation of all instruments and to remove erroneous measurements. Additionally, a quality flag has been added to the data sets to communicate the quality of each data point. The data have been regridded onto a unified grid for easy combined analyses of the measurements. The HAMP data, together with dropsonde measurements, provide insights into cloud geometry, hydrometeors, vertically integrated humidity

and thermodynamic profiles.



Section 2 gives specific information about the four campaigns. A detailed description of the aircraft and instruments is given in Sect. 3. The data set is described in Sect. 4 and the quality control is outlined in Sect. 5. Information about how the data can be accessed is given in Sect. 6.

## 2 Campaigns with NARVAL payload

The HAMP instrument suite so far has been part of the HALO payload (refered to as the NARVAL payload) during four campaigns: NARVAL1 (with sub-campaigns NARVAL-South and NARVAL-North), NARVAL2 and NAWDEX. Two campaigns focused on the observation of clouds and convection over the tropical Atlantic during dry season (NARVAL-South) and wet season (NARVAL2). The goals of the other two campaigns were the observation of clouds and convection associated with cold sector (NARVAL-North) and warm sector (NAWDEX) of mid-latitude cyclones. During all campaigns, all transfer flights were
also research flights with full measurements. The individual campaigns are described in more detail below.

### 2.1 NARVAL-South

The first NARVAL campaign (NARVAL-South) took place between 10 December 2013 and 22 December 2013. This campaign and the directly following NARVAL-North campaign were first and foremost planned as a demonstrator mission to assess the capabilities of HALO and HAMP to work as an airborne cloud remote sensing platform. Additionally, the focus of this cam-
paign was to assess the representativeness of ground-based measurements on Barbados (Stevens et al., 2016) to a broader trade wind region over the tropical Atlantic, and to enable satellite observation evaluation by collocating tracks (Klepp et al., 2014; Stevens et al., 2016). In total, eight flights amounting to about 67 flight hours were conducted. Of the eight flights, four flights were cross-Atlantic transects during transfers between Oberpfaffenhofen, Germany (OBF) and Grantley Adams International Airport, Barbados (BGI) and four were local flights over the tropical North Atlantic east (and upwind) of Barbados. The overall
flight time for individual flights was around 10 hours for transatlantic flights and around 7 hours for local flights. Target areas of this campaign were the trade wind region east of Barbados as well as cross-Atlantic transects. The synoptic situation was relatively constant from flight to flight. Therefore, flight planning was done mainly in the interest of gathering statistically sound data and to compare with satellite data. A-Train collocations were achieved during seven flights. All flights and their aims are listed in Table 1 and flight tracks are shown in Fig. 1 (a).

### 2.2 NARVAL-North

Following directly after the first campaign, NARVAL-North took place from 07 January 2014 to 22 January 2014. Center of operation was at Keflavik International Airport, Iceland (KEF). During this campaign, seven flights were conducted, overall amounting to 46 flight hours. The target area were the extra-tropical North Atlantic south and south-west of Iceland. The aim of NARVAL-North was to analyze post-frontal convective regimes of mid-latitude cyclones over the extra-tropical North
Atlantic and to investigate the accuracy of existing satellite precipitation climatologies (Klepp et al., 2014). Flight planning was done taking into account current synoptic situations. The flights mainly sampled the cold sector of mid-latitude cyclones



as well as some occlusions. A-Train collocations were achieved during four flights. Figure 1 (b) shows the flight tracks of NARVAL-North and in Table 2 all flights and their aims are listed.

## 2.3 NARVAL2

NARVAL2 followed up on NARVAL-South with a stronger focus on the local region around Barbados and over the tropical
Atlantic between 8 August 2016 and 30 August 2016. Ten research flights were flown in total of about 84 flight hours (Table 3) with different flight tracks (Fig. 1 (c)). Center of operation was again BGI on Barbados. A-Train collocations were achieved during five flights, as well as collocations with the satellites GPM (Global Precipitation Measurement) and Megha-Tropiques. The aim of the campaign was to assess the interaction between large-scale dynamics and evolution of convective systems over the tropical Atlantic (Stevens et al., 2018). Of the 10 flights during this campaign, 2 flights were transfers to and from Barbados.
The remaining flights took place over the tropical North Atlantic east and south-east of Barbados. Three of these flights focused on clouds in and associated with the Intertropical Convergence Zone (ITCZ). The other five flights took place further north and mainly over shallow convection. Two flights focused on the surrounding atmosphere of Hurricane Gaston (2016). However, due to instrument failure, no radar measurements are available from these two flights. A special flight pattern, i.e. circles with a radius of about 110 km with frequent dropsonde launches was performed in order to measure the large-scale vertical motion
(Bony and Stevens, 2018) on six flights.

## 2.4 NAWDEX

The fourth campaign of this set, NAWDEX took place from 17 September 2016 until 18 October 2016. 13 flights amounted to about 96 flight hours. During this campaign, HALO, along with other research aircraft, was stationed at KEF on Iceland. Figure 1 (d) shows the flight tracks of NAWDEX. A-Train collocation was achieved during one flight. Of the 13 flights of
this campaign, 2 were transfers to and from Keflavik. The main target of the flights during this campaign were the clouds associated with the warm conveyor belt of mid-latitude cyclones (Schäfler et al., 2018). Flight planning was done taking into account current synoptic situations. In Table 4, all flights and their aims are listed. The flights mainly sampled the warm sector and frontal systems of mid-latitude cyclones.

## 3 Aircraft and instrumentation

Instruments described in this paper were part of the NARVAL payload on board the German research aircraft HALO (High Altitude LOng range research aircraft). The aircraft, the HAMP instruments, and the dropsondes are described in the following sections.

### 3.1 The HALO aircraft

The HALO aircraft is a Gulfstream G550 with ceiling altitude up to 15 km and long endurance of up to 10 h (Krautstrunk
and Giez, 2012; Wendisch et al., 2016). HALO's capabilities of high ceiling and long range enables researchers to cover a



wide range of atmospheric conditions during a single flight and to fly above main airline traffic and most cloud systems. Thus, measurements on board HALO often provide a view of at least a large part of or even the entire troposphere by measurements. The median flight altitude was 13.1 km for the NARVAL-South campaign and 7.8 km during NARVAL-North taking into account the lower tropopause height of the target region in winter and flight restrictions. During NARVAL2, median flight

altitude was 9.7 km and during NAWDEX 12.4 km. The lower altitudes during NARVAL2 were flown to accommodate for the fact that dropsondes were launched in quick succession during some of these flights (see Sect. 3.2.3).

The HALO aircraft is equipped with the BAsic HALO Measurement And Sensor system (BAHAMAS). This instrument system provides data about the aircraft's location and attitude (position, altitude, heading, roll and pitch angle) and atmospheric measurements at aircraft level (e.g. temperature, humidity, pressure) (Krautstrunk and Giez, 2012). During the first two

campaigns, data were sampled with 1 Hz, in 2016 also 100 Hz data were available. In this data set, however, the 1 Hz data are provided for all flights, since these correspond best to the sampling rates of the other instruments. Measurements of aircraft location (latitude, longitude, altitude above WGS84 ellipsoid) and attitude (angles of roll, pitch, and heading) are provided in this data set. Note, that most measurements were performed in straight legs or large circles with only 6.72 % of the measurements having a roll angle of more than $5°$.

The median speed above ground was $224 \, \mathrm{m \, s^{-1}}$ and therefore measurements with a resolution of 1 s roughly represent a distance of about 200 m. Typically, the time from start to reaching flight level amounted to 35 minutes and the final descent to 30 minutes.

## 3.2 Instrument description

The HALO Microwave Package (HAMP, Mech et al., 2014) comprises microwave radiometers with 26 channels in the range

between 20 and 183 GHz and a 35 GHz cloud radar. The radiometer modules and radar antenna are mounted below the fuselage inside the belly pod (see Fig. 1 of Mech et al. (2014)) with nadir viewing direction. Mech et al. (2014) illustrate the sensitivity of the different measurements with respect to hydrometeors. Besides the HAMP data, also profiles from dropsonde measurements and auxiliary data from the aircraft's basic data system are included in the published data set described here. Specifications of these instruments are listed in Table 5 and are described in the following sections.

### 25 3.2.1 Microwave radiometers

The HAMP microwave radiometers were custom-manufactured by Radiometer Physics GmbH (RPG, Meckenheim, Germany). They comprise five modules which measure time series of brightness temperatures at 26 frequencies with a sampling rate of approximately 1 Hz. The 22 GHz and the 183 GHz rotational water vapor lines are probed each with seven channels along the line. In contrast to the 22 GHz module, the 183 GHz hosts a double sideband receiver and therefore the signal is a composite

from passbands on both sides of the line. The oxygen absorption complex at 60 GHz is measured with seven single sideband channels while the double sideband receiver of the 118 GHz oxygen line has four channels. In addition, a window channel at 90 GHz is highly sensitive to liquid water emission. The noise-equivalent delta temperature (NeDT) has been determined for inflight scenes and is best for the lowest frequencies (below 0.3 K) and worst for the 90 GHz channel and 183 GHz bank




(below 0.6 K). Mech et al. (2014) provide more details on instrument specifications. The receivers undergo frequent relative calibration during flight and therefore the most critical features of the microwave radiometer measurements is the absolute calibration (cf. Küchler et al., 2016) which is discussed further in Sect. 5.1.

### 3.2.2 Cloud radar

The MIRA35 cloud radar was manufactured by METEK GmbH (Elmshorn, Germany). It operates in the Ka-band at 35 GHz and is a monostatic, pulsed, magnetron, Doppler radar (Mech et al., 2014). An advantage of this frequency over the often used W-band is that it is less affected by attenuation due to condensate. Measured variables are mainly profiles of reflectivity, linear depolarization ratio, Doppler spectra and Doppler velocity. In case of the HAMP radar, measurement of Doppler velocity was strongly affected by aircraft motion and is therefore not provided. In this data set reflectivity and linear depolarization ratio are

provided. Data sampling rate is 1 Hz. At 13 km flight altitude, sensitivity is $\approx$ -30 dBZ and footprint size is $\approx$ 130 m. Vertical resolution of the measurements is 28.8 m. Only measurements above approximately 6 km flight altitude were conducted to avoid a too strong return from the surface into the receiver at lower levels. After some initial analyses a offset was discovered and quantified (Ewald et al., 2018). This correction is discussed further in Sect. 5.3.

### 3.2.3 Dropsondes

Vaisala RD94 dropsondes (Busen, 2012) were used in all campaigns. Overall, 525 dropsondes were released. These sondes were released from the aircraft and took measurements of profiles of temperature, humidity, pressure, wind speed, wind direction, and location, while falling to the surface. Data sampling rate of these sondes was usually 2 Hz. The locations for releasing the dropsondes were chosen based on the atmospheric conditions. We attempted to deploy at least one sonde in clear sky condition to have as a reference for radiometer retrievals. For most of the flights, at least one dropsonde per flight was released.

Median drift length of dropsondes during their descent was 3.8 km (lower quartile 2.3 km and upper quartile 10.8 km).

Operational constraints limited where sondes could be released. Air traffic control had to clear every release in advance. This request was only granted if the airspace below the research aircraft was empty. Therefore, it was not always possible to release the sondes were it was most meaningful from a meteorological point of view. During the northern campaigns, we often had to decide to either fly below the transatlantic air traffic and be able to release sondes or above the traffic to get a more

comprehensive profile of the entire troposphere without releasing sondes.

Another constraint for dropsondes was that the receiving unit on board the aircraft could record measurements from up to four dropsondes simultaneously. So, to launch sondes in quick succession, as was necessary for some of the NARVAL2 flights, and still record the entire profile, the decision was made to fly at lower altitudes of about 8 km during flights where this was necessary.



## 4  Data description

An in-depth description of the principles of HAMP radar and radiometer measurements is given in Mech et al. (2014). Here, only an example will be discussed to give an overview of how the measurements can be interpreted.

A combined snapshot of radar and radiometer measurements is shown in Fig. 2. The four top panels with time series of
brightness temperatures $T_b$ from different radiometer modules show integrated information of the atmosphere below the aircraft at each time. Over the radiatively cold ocean, the emission by liquid water can be seen as an increase in $T_b$. The strength of the emission increases with frequency and can be best seen in the window channels, at 31.4, 90 and $118 \pm 8.5$ GHz, (Fig. 2 (a), (c)) that are less affected by emission from water vapour and oxygen. In combination with channels sensitive to water vapor, e.g. 22 GHz (Fig. 2 (a)), the liquid water path can be derived using statistical algorithms (cf. Schnitt et al., 2017). Scattering
of microwave radiation on ice particles strongly increases in strength with increasing frequency. The strong $T_b$ depressions in channels around 183 GHz (Fig. 2 (d)) result from scattering by larger ice particles and can be used to infer ice content. Channels along the 60 GHz oxygen complex (Fig. 2 (b)) can penetrate the atmosphere the more the further away they are from the absorption maximum and can in this way give information on the temperature profile. Combinations of measurements from different channels can be used to derive temperature and humidity profiles, as well as liquid/snow water path. In summary, the
combination of measurements from different channels can be used to derive the integrated water vapor and liquid water path (Schnitt et al., 2017), coarse resolution temperature and moisture profiles as well as information on ice and snow occurrence.

The radar reflectivity (Fig. 2 (e)) shows a cross section of the clouds the aircraft passed over. Reflectivity of the cloud radar at 35 GHz is a measure of size and number of cloud droplets. The linear depolarization ratio (LDR, Fig. 2 (f)) is defined as ratio of cross-polarized reflectivity factor to co-polarized reflectivity factor. For a perfectly spherical particle, the backscattering in
the cross-polarized reflectivity factor is zero in linear units and $-\infty$ in logarithmic dB units and thus the ratio also is zero or $-\infty$, respectively. The LDR increases the more the shape of the scattering particle deviates from perfect symmetry (Oue et al., 2015). In Fig. 2 (e) and (f), the melting layer is visible as a horizontal line of high reflectivity together with high LDR values just below 1 km.

## 5  Quality control and data processing

Data processing has been done to convert the measured data into an easily usable format and to ensure good quality of the data. The data have been inspected and flagged accordingly. Finally, all data have been transformed onto a unified grid with a temporal resolution of 1 s. For cloud radar and dropsondes the vertical resolution is 30 m. These steps will be briefly discussed below.

### 5.1  Radiometer calibration

The radiometers were calibrated on the ground before almost each flight using the manufacturer's warm and cold load calibration method. In this method each radiometer was pointed successively on a warm black-body target at ambient temperature and



on a cold black-body target cooled down to the boiling point of liquid nitrogen (LN$_2$). Targets with an open air-LN$_2$ interface were used during the NARVAL1 campaigns. Later, these were exchanged with targets embedded in a foam box that is transparent to microwaves avoiding reflections at the LN$_2$ interface. The new targets were vertically oriented and a metal mirror was used to redirect the view from nadir to horizontal. During flight, the radiometers were continuously calibrated using two

reference loads. More details are given by Mech et al. (2014).

The quality of the original brightness temperature measurements was evaluated by comparing them with synthetic ones simulated from dropsonde profiles. For this comparison, the sondes were filtered for clear sky conditions with low atmospheric variability and their thermodynamic profiles were fed into the Passive and Active Microwave TRAnsfer model (PAMTRA). Random discrepancies between both can be attributed to the matching of HAMP nadir measurement and the drifting sondes.

The systematic differences could be identified which cannot be explained by systematic errors of the drop sonde measurement nor the microwave absorption model. Most likely they result from changes within the bellypod during take-off. Because some differences in the biases between different flights could be found, a bias correction based on the mean differences between synthetic and measured brightness temperatures was performed. However, for some flights only very few or no drop sondes are available. In order to arrive at robust correction, those flights having three or less dropsondes were corrected using the

campaign mean. For information, the offset corrections used for each channel are included in the data files.

## 5.2   Removal of erroneous radiometer measurements

Some obvious errors in radiometer measurements occurred during the campaigns. During the NARVAL-South and NARVAL2 campaigns, the 183 GHz radiometer module suffered from instabilities during the initial phase of some flights. Because of strong drifts, the regular gain calibration (performed around every five minutes) caused a saw tooth pattern whose amplitude

reduced with time as the module became more stable. After NARVAL2, a broken dielectric resonator oscillator (DRO) was replaced. Furthermore, at some instances unreasonably high or low values were recorded by different modules likely due to instabilities in data transfer. This is also thought to be the reason that time stamps sometimes did not increase continuously but jumped ahead or backwards in time.

All data were inspected thoroughly by eye and errors of brightness temperature from saw tooth patterns or spikes in data were

not corrected but removed from the data set. It was attempted to reconstruct the erroneous time stamps where possible. If the interval with faulty time stamps was too long and reconstruction was not possible, the data have been removed. Additionally, also measurements during turns (roll angle $> 5°$) or when the aircraft was below 6 km altitude have been removed.

## 5.3   Reflectivity bias correction

By comparison with airborne and spaceborne radar measurements at 95 GHz, it has been observed that HAMP radar reflec-

tivity appeared to be too low by about about 8–10 dBZ. To assess the exact value of this offset, Ewald et al. (2018) derived a calibration value for the HAMP cloud radar reflectivity. To this end, calibration maneuvers with constant bank were flown during NARVAL2 and NAWDEX while the radar emitting power was reduced to use the returned sea surface signal. Additionally, individual instrument components were measured to calibrate the cloud radar. This calibration was validated with



other airborne and spaceborne measurements. They concluded that the resulting bias of 7.6 dBZ originated from differences in software configuration and instrument calibration. This value is constant over the entire measurement range and has been added to all reflectivity measurements in this data set.

### 5.4 Radar data quality flag

The radar data include some features that might not be desirable to use: in the beginning of flights and sometimes also around measurement interruptions when data were recorded but no radiation was emitted, the data include noise. Further, on a couple of flights, calibration maneuvers for the radar have been executed (Sect. 5.3). During these maneuvers, the transmitting power of the radar has been changed and thus the received signal also changed. Additionally, especially when the aircraft overflew land, the unified radar data contains also pixels that are at or below the surface. All these cases – surface and sub-surface, sea

surface, noise, intervals with calibration maneuvers – have been marked in an additional data flag variable indicating the state of radar data: data ok (0), noise (1), surface or sub-surface (2), sea surface (3), radar calibration (4). For additional information, a variable that indicates turning of the aircraft (roll angle $> 5°$) has been added to the data set.

### 5.5 Temporal collocation

To ensure comparability of measurements of a certain time interval, a good temporal collocation of the instruments is needed.

Temporal collocation of HAMP measurements with HALO's on-board instrumentation (BAHAMAS) has been performed independently for radar and microwave radiometers. A good synchronization with BAHAMAS lets us then assume that also radar and radiometer measurements are synchronized well.

Temporal collocation between cloud radar and BAHAMAS has been checked by using the aircraft attitude data from BAHAMAS to correct the radar data for this attitude. The idea is that, if time stamps between both systems match perfectly,

the surface return signal from the ocean surface should be a straight line, even during turns of the aircraft. To test for the time difference, the time series have been shifted and the quality of attitude correction has been assessed by looking at the variance of height of the surface (taken as the maximum signal in each profile). The accuracy of this procedure is estimated to be 2 s. The analyzed temporal differences mainly ranged between 0 s and 2 s. The maximal difference was 19 s. Measurements were corrected with these found offsets.

Temporal collocation between radiometer modules and BAHAMAS has been investigated by looking at the transition between land and sea. This happens shortly after take-off or before landing since both airports, Barbados Grantley Adams Airport and Keflavik Airport, are located close to the coast. This becomes possible as the microwave emissivity at low frequencies strongly differs between ocean (about 0.5) and land (about 0.9). The NAVO/GHRSST global 1 km land sea mask (https://www.ghrsst.org/ghrsst-data-services/tools/) has been used to identify locations where the aircraft passed the shore. The

footprint of the radiometers at an altitude of 900 m during take-off is approximately 70 m long which corresponds to 0.7 s at groundspeed of $105 \, \text{m s}^{-1}$. The aircraft is flying with a pitch of about 11 degree and therefore the instruments look ahead of the aircraft position. Due to this "looking ahead" and the calculated footprint at this altitude, the accuracy of the estimated land-sea-transition is about 2.5 s. The land sea mask resolution is approximatley $0.0083°$. Using this, the crossing of the shore



can be determined within 3 to 12 s, depending on position (0.0083° longitude is less distance in the mid-latitudes than it is in the tropics) and aircraft speed. After review of the measurements, all time series stayed in this interval and are thus deemed correct.

## 5.6 Regridding

As mentioned in Sect. 3 and Table 5, the instruments have different sampling rates and in case of radar and dropsondes also different vertical resolutions. To create a self-contained data set where variables from different instruments are easily comparable in time and height, the data have been transformed onto a uniform grid for each flight.

Radar measurements have additionally been corrected for aircraft attitude. After this correction, the vertical coordinate no longer corresponding to range from instrument, but to height above surface. This makes the vertical profiles of radar data from different flights comparable with each other. After that, all data have been interpolated to the new grid with 30 m vertical and 1 s temporal resolution using the nearest neighbor value. The resulting data set consists of radar data with 1 s temporal and 30 m vertical resolution, radiometer data with 1 s temporal resolution, and dropsonde data with 30 m vertical resolution.

## 5.7 Filtering and gap filling

In addition to the quality control described in Sect. 5.2, spikes in dropsonde and BAHAMAS data have been filtered out. Spikes in dropsonde profiles were identified as jumps in data between two measurements by more than half of the data range of the entire profile. Spikes in BAHAMAS data have been identified by eye. Data gaps in dropsonde profiles have been interpolated if the gap was shorter than 10 s. With an average falling velocity of about $12\,\mathrm{m\,s^{-1}}$, this corresponds to roughly 120 m. BAHAMAS and radiometer time series have been interpolated if the gap was not longer than 3000 s (BAHAMAS) and 30 s (radiometers), which at average aircraft velocity of $200\,\mathrm{m\,s^{-1}}$ corresponds to 6,000 km and 6 km respectively. The reasoning behind these different thresholds is that BAHAMAS data fluctuate very little in the upper troposphere/lower stratosphere. Therefore, it is reasonable to interpolate longer intervals than for radiometer data. A flag has been added to radiometer data, indicating which values were interpolated.

## 6 Data availability

The data for all flights described here are submitted to the CERA database (https://cera-www.dkrz.de/WDCC/ui/cerasearch/). The data have been released under Creative Commons Attribution-NonCommercial-ShareAlike 4.0 (CC BY-NC-SA 4.0). The data are saved as separate files for radiometer, radar, and dropsonde measurements. Radiometer and radar data sets contain auxiliary data (aircraft location and attitude). Data from each flight are saved into individual files. The data format is NetCDF (version 4).

Along with the data, quicklooks for all flights have been uploaded to the CERA database as auxiliary data. Figure 3 shows an example for one entire flight. These figures include time series of radar reflectivity, microwave radiometer brightness temperatures, aircraft attitude, and the flight track and give a quick overview of the measurement from each flight.





The data are structured into four sets according to the four campaigns. Each set is associated with one digital object identifier: NARVAL-South (https://doi.org/10.1594/WDCC/HALO_measurements_2), NARVAL-North (https://doi.org/10.1594/WDCC/HALO_measurements_1), NARVAL2 (https://doi.org/10.1594/WDCC/HALO_measurements_3), NAWDEX (https://doi.org/10.1594/WDCC/HALO_measurements_4).

## 7 Summary

The HAlo Microwave Package (HAMP) was deployed on the German research aircraft HALO (High Altitude LOng range research aircraft) during four campaigns. The four campaigns took place between December 2013 and October 2016 out of Barbados and Iceland. Measured situations cover dry and wet season over the tropical Atlantic and cold and warm sector of mid-latitude cyclones. HAMP comprises microwave radiometers with 26 channels in the range between 20 and 183 GHz and a 35 GHz cloud radar. In sum, measurements were recorded during 295 flight hours. The flights covered data over a variety of atmospheric conditions but enough of similar conditions to ensure statistically representative samples. Measurements of cloud radar reflectivity and linear depolarization ratio at 35 GHz, radiometer brightness temperatures between 20 and 183 GHz and dropsonde atmospheric profiles are provided in the data set described here. Quality control has been performed to remove outliers and ensure temporal collocation of the instruments. The data has been regridded onto a uniform grid for easy combined analyses. The data set has been submitted to the CERA database (https://cera-www.dkrz.de/WDCC/ui/cerasearch/, https://doi.org/10.1594/WDCC/HALO_measurements_1, https://doi.org/10.1594/WDCC/HALO_measurements_2, https://doi.org/10.1594/WDCC/HALO_measurements_3, https://doi.org/10.1594/WDCC/HALO_measurements_4) for free access. This data set adds to the relatively sparse observations of maritime clouds in the tropics and mid-latitudes. The data allow for analyses to get insight into cloud properties and atmospheric state.

Further flights with HAMP will be performed in January 2020 as part of the EUREC[4]A campaign (Bony et al., 2017) and in March 2021 as part of the HALO-(AC)[3] campaign.

*Author contributions.* FA, SC, and BS were initiators of the HAMP project. LH, FJ, MH, and MM carried out initial planning and installation of the HAMP instrument suite. FE, LH, and MH recalibrated the radar reflectivity data. SC, MJ, and MM derived the calibration for radiometer brightness temperatures. HK carried out quality control of the data, developed the unified data set of the different instruments, and wrote the paper with support and input from all co-authors.

*Competing interests.* The authors declare that they have no conflict of interest.

*Acknowledgements.* This work was supported by the German Research Foundation (Deutsche Forschungsgemeinschaft, DFG Priority Program SPP 1294) and by the Max Planck Society. The NAWDEX campaign was additionally funded within SFB/TRR165 Waves to Weather.



Thanks to the entire NAWDEX communitiy for the great collaboration. Operating the HAMP instruments during four campaigns was a team effort. The authors would like to thank all additional operators who helped calibrating and running the instruments during the campaigns: Stephan Bakan, Björn Brüggmann, Lisa Dirks, Akio Hansen, David Hellmann, Christian Klepp, Marcus Klingebiel, Emiliano Orlandi. Katharina Bayer is thanked for her great help in preparing the data for publication.



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





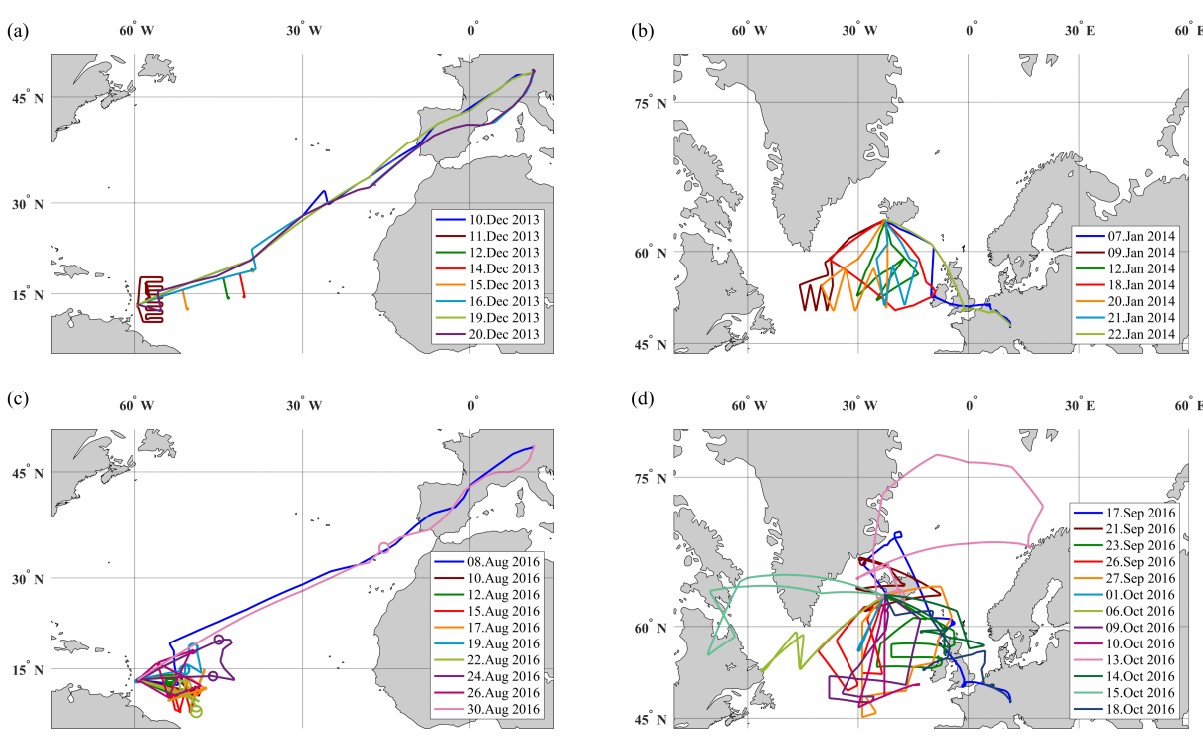

**Figure 1.** Flight tracks from all four campaigns: NARVAL-South (a), NARVAL-North (b), NARVAL2 (c), NAWDEX (d)





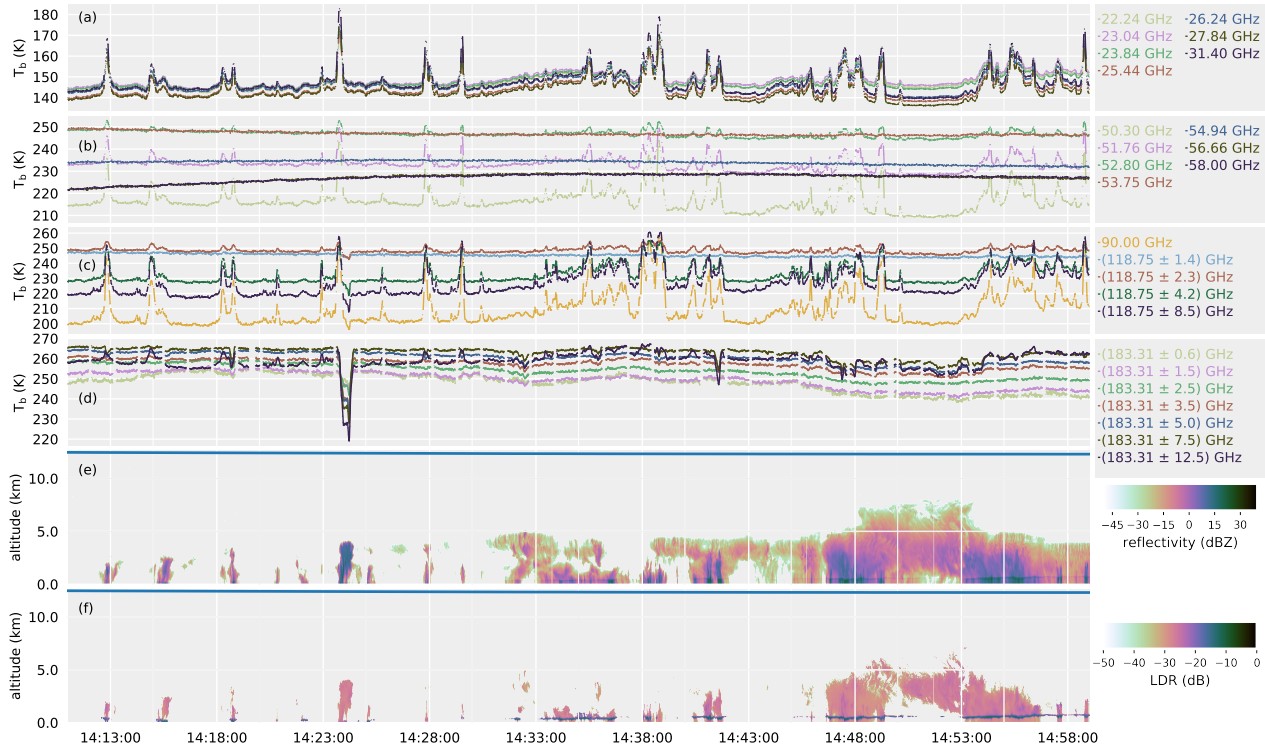

**Figure 2.** Quicklook of HAMP measurements during NAWDEX flight on 06 Oct 2016. (a-d) Time series of brightness temperatures (lines) for individual radiometer modules. (e) Profiles of radar reflectivity (shaded) and HALO's flight altitude (solid line). (f) Profiles of radar linear depolarization ratio and HALO's flight altitude (solid line). The distance traveled by the aircraft during this time is roughly 710 km.

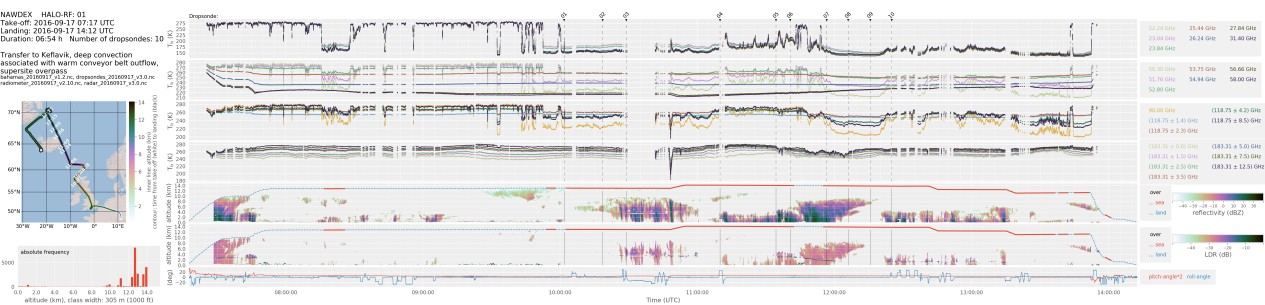

**Figure 3.** Example of quicklook of HAMP measurements that are available along with the data set in the CERA database. Left panel (top to bottom): flight information, flight track, altitude histogram. Center top four panels (lines): microwave radiometer brightness temperatures; fifth panel (colors): cloud radar reflectivity; sixth panel (colors): cloud radar linear depolarization ratio; bottom panel (lines): aircraft attitude. Vertical lines in center panel denote times of dropsondes released.





**Table 1.** List of NARVAL-South flights.

| Date (UTC) | Start time (UTC) | End time (UTC) | Duration (hh:mm) | Number of dropsondes | Median altitude (km) | Flight characteristics |
|---|---|---|---|---|---|---|
| 10 Dec 2013 | 10:13 | 20:41 | 10:27 | 14 | 12.8 | shallow trade wind convection, transatlantic cross section, A-Train overpass |
| 11 Dec 2013 | 14:28 | 21:58 | 7:29 | 6 | 13.1 | shallow trade wind convection, local flight with mattress pattern, A-Train overpass[+] |
| 12 Dec 2013 | 13:49 | 20:20 | 6:30 | 9 | 13.1 | shallow trade wind convection, mid-Atlantic cross section, A-Train overpass[+] |
| 14 Dec 2013 | 13:35 | 20:21 | 6:45 | 11 | 13.2 | shallow trade wind convection, mid-Atlantic cross section, A-Train overpass |
| 15 Dec 2013 | 15:15 | 21:45 | 6:30 | 9 | 13.2 | shallow trade wind convection, mid-Atlantic cross section, A-Train overpass |
| 16 Dec 2013 | 13:10 | 22:59 | 9:48 | 10 | 13.2 | shallow trade wind convection, transatlantic cross section, A-Train overpass |
| 19 Dec 2013 | 10:06 | 19:56 | 9:49 | 9 | 13.6 | shallow trade wind convection, deep convection over central Atlantic, transatlantic cross section, collocation with French Falcon |
| 20 Dec 2013 | 16:30 | 02:32 | 10:02 | 8 | 13.1 | shallow trade wind convection, deep convection over central Atlantic, transatlantic cross section, A-Train overpass |

[+] 183 GHz radiometer module working only part of the time



**Table 2.** List of NARVAL-North flights.

| Date (UTC) | Start time (UTC) | End time (UTC) | Duration (hh:mm) | Number of dropsondes | Median altitude (km) | Flight characteristics |
|---|---|---|---|---|---|---|
| 07 Jan 2014 | 12:07 | 17:49 | 5:42 | 0 | 12.1 | transfer to Keflavik, deep convection, supersite overpasses |
| 09 Jan 2014 | 08:14 | 17:20 | 9:06 | 11 | 7.3 | mid level clouds in cold air outbreak, zigzag pattern, post-frontal low, A-Train overpass |
| 12 Jan 2014 | 08:32 | 15:11 | 6:38 | 12 | 7.7 | zigzag pattern through core and occlusion of mature mid-latitude cyclone, A-Train overpass |
| 18 Jan 2014 | 08:56 | 14:49 | 5:53 | 5 | 7.9 | convective development from shallow to deep in cold air behind cyclone, box pattern with altitude change, A-Train overpass |
| 20 Jan 2014 | 10:16 | 18:45 | 8:29 | 11 | 7.8 | convection in core of weak cyclone, zigzag pattern across system |
| 21 Jan 2014 | 10:51 | 17:00 | 6:08 | 7 | 7.8 | re-intensified cold air convection, zigzag pattern across system, A-Train overpass |
| 22 Jan 2014 | 10:01 | 14:26 | 4:24 | 0 | 12.2 | transfer from Keflavik, deep convection, Supersite overpasses |



**Table 3.** List of NARVAL2 flights.

| Date (UTC) | Start time (UTC) | End time (UTC) | Duration (hh:mm) | Number of dropsondes | Median altitude (km) | Flight characteristics |
|---|---|---|---|---|---|---|
| 08 Aug 2016 | 08:13 | 18:51 | 10:38 | 9 | 14.5 | transatlantic transfer, A-Train overpass[+] |
| 10 Aug 2016 | 11:52 | 20:07 | 8:15 | 30 | 8.1 | dry air to deep convection at the edge of the ITCZ, large circles with connecting mattress pattern, A-Train overpass[+] |
| 12 Aug 2016 | 11:43 | 19:37 | 7:53 | 50 | 9.7 | dry air with few shallow clouds, large circles with connecting mattress pattern[+] |
| 15 Aug 2016 | 11:48 | 19:45 | 7:57 | 10 | 9.7 | shallow to deep convection while crossing in and out of the ITCZ, zigzag pattern, A-Train overpass[+] |
| 17 Aug 2016 | 14:48 | 23:07 | 8:19 | 12 | 14.4 | shallow convection and high clouds in moist air, straight legs for satellite overpasses, collocation with A-Train, Megha-Tropiques, GPM satellites[+] |
| 19 Aug 2016 | 12:29 | 20:52 | 8:23 | 50 | 9.7 | partly dust laden and cloud free air, partly clearer air with shallow convection, large circles, A-Train and Megha-Tropiques overpass[+] |
| 22 Aug 2016 | 13:16 | 20:57 | 7:40 | 13 | 9.7 | deep convection associated with ITCZ, large circles[+] |
| 24 Aug 2016 | 12:43 | 20:54 | 8:10 | 12 | 9.7 | dry inflow region of hurricane Gaston (2016), deep convection at edge of hurricane, large circles and arch along hurricane[*+] |
| 26 Aug 2016 | 13:43 | 20:54 | 7:10 | 12 | 9.0 | Gradients of air masses: dry, shallow convection to convection in moist air, Saharan dust layer, cross pattern[*+] |
| 30 Aug 2016 | 09:42 | 19:52 | 10:09 | 17 | 13.9 | transatlantic transfer[*+] |

[*] no radar measurements; [+] 183 GHz radiometer module working only part of the time



**Table 4.** List of NAWDEX flights.

| Date (UTC) | Start time (UTC) | End time (UTC) | Duration (hh:mm) | Number of dropsondes | Median altitude (km) | Flight characteristics |
|---|---|---|---|---|---|---|
| 17 Sep 2016 | 07:17 | 14:12 | 6:54 | 10 | 12.9 | transfer to Keflavik, deep convection associated with warm conveyor belt outflow, supersite overpass |
| 21 Sep 2016 | 13:55 | 19:25 | 5:29 | 14 | 12.9 | deep convection associated with warm conveyor belt ascent and outflow region, legs across frontal region, GPM collocation |
| 23 Sep 2016 | 07:36 | 16:36 | 9:00 | 21 | 12.0 | deep convection associated with warm conveyor belt ascent and outflow region, straight legs across frontal region |
| 26 Sep 2016 | 09:57 | 18:59 | 9:02 | 25 | 9.0 | deep convection associated with warm conveyor belt ascent region, straight legs across cyclone center and frontal region |
| 27 Sep 2016 | 11:32 | 20:37 | 9:05 | 20 | 9.0 | cold air sector, strong moisture transport in warm sector, box pattern across cold and warm sector |
| 06 Oct 2016 | 07:02 | 16:13 | 9:10 | 20 | 8.4 | horizontal and vertical moisture gradients across a tropopause polar vortex, zigzag across moisture gradients |
| 09 Oct 2016 | 10:24 | 19:04 | 8:39 | 1 | 13.3 | shallow convection in cold sector, deep convection in warm conveyor belt ascent region, straight legs through warm conveyor belt |
| 10 Oct 2016 | 11:58 | 19:37 | 7:38 | 19 | 8.7 | warm conveyor belt ascent and outflow region, warm sector, cold sector, and core of mid-latitude cyclone, straight legs through core and frontal region of cyclone |
| 13 Oct 2016 | 07:58 | 15:58 | 7:59 | 24 | 13.2 | warm conveyor belt along ridge, supersite overpass, half circle around ridge from Greenland, south of Svalbard to Norway |
| 14 Oct 2016 | 08:23 | 14:53 | 6:29 | 7 | 12.5 | aircraft collocation with French Falcon and FAAM BAE, straight legs for aircraft collocations, A-Train overpass |
| 15 Oct 2016 | 08:41 | 16:36 | 7:55 | 12 | 12.3 | shallow to mid-level clouds over Labrador sea, straight legs across frontal zone, zigzag across cold air |
| 18 Oct 2016 | 08:51 | 14:41 | 5:50 | 15 | 13.1 | transfer from Keflavik, mid level and deep convection, supersite overpass |




**Table 5.** Instruments used all four campaigns that provided measurements for this data set.

| Name | Instrument | Measured quantities | Sampling frequency |
|---|---|---|---|
| HAMP radiometers | microwave radiometers @ K-band (22 - 31 GHz), V-band (50 - 58 GHz), W-band (90 GHz), F-band(119 GHz) and G-band (183 GHz) | brightness temperatures | ≈1 Hz |
| HAMP cloud radar | Ka band (35 GHz) pulsed magnetron radar | profiles of radar reflectivity, depol. ratio, Doppler velocity | 1 Hz |
| dropsondes | AVAPS receiver using VAISALLA RD-94 | profiles of relative humidity, temperature, horizontal wind | 1 Hz |
| BAHAMAS | | aircraft attitude and location | 100 Hz, 1 Hz |