# Peer review of "A unified data set of airborne cloud remote sensing using the HALO Microwave Package (HAMP)"

_Earth System Science Data, 2018_

## Referee Comment (RC1) · Anonymous Referee #1 · 26 Jan 2019

Review of: A unified dataset of airborne cloud remote sensing using the HALO Microwave Package (HAMP)

document number: essd-2018-116

Accept with minor revisions

This manuscript provides an overview of the HAMP data gathered from 37 flights of 4 campaigns. The datasets have been made publicly-available, with the DOIs included in the abstract.  The manuscript is well organized and the datasets are adequately described. The included links to the datasets are functionnal. This is the appropriate journal for this type of manuscript.

My main 3 comments are that 1) it would be nice to see an example of at least one scientific application of the dataset, including retrieved liquid water path, water vapor paths, radar/lidar derived cloud properties. Figures 2 and 3 are good as it is also nice to see what the actual quicklooks look like. These also communicate that the retrieval work remains to be done. Are there any examples of retrieved physical quantities from any of the flights that could be mentioned? Or any scientific results, perhaps already published in other papers? This could be considered to fall outside the scope of the ESSD journal, and if the editor also feels that way, then so be it, but it would help increase the impact of the manuscript to show the reader why they might be interested in working with it. My second comment is that the manuscript contains many small grammatical inconsistencies. These merely reflect that the authors are not native English speakers and the meaning is generally clear, but if a native English speaker could be found to read through the manuscipt and improve the English, that would be worth doing. My third comment is that the description of the Vaisala dropsondes, "RD94" on p. 6, doesn't mean anything to mean. Some description and a reference to their performance would be nice. Is a correction incorporated for a solar radiation bias, as is done for research grade radiosondes in the US? Are there any issues for the user to be aware of?

Minor comments:

page 1, line 21: "the full atmospheric column…." this depends on the frequency, would be worth stating.

page 2, line 4-5: "12 dBZ….typical for light precip.." is confusing. How about : "12 dBZ, more suited for the detection of heavier precipitation, and less so for characterizing shallow convection."

p. 2, line 6: to -> with

p. 2 line 22: put "The" before "Center"
line 24: put 'was' before 'again'
lines 27,28: the verb tenses are not consistent.

p. 3, line 16: "collocating tracks": the word 'collocating' is used a lot within the manuscript. This is the first instance. At first I was not sure what the authors meant, it is not a typical usage. Perhaps rephrase as "and to evaluate satellite observations through flying underneath the

satellite,  along the projected satellite track near the satellite overpass time". Once this is clearly defined the continued use of 'collocated' will be clear I think.
line 28: were -> was

p. 4, line 5: in -> for a
line 6: always preceed "Center" with a "The".

p.5, line 10: with -> at, in -> while

p.6
line 20: HOw can the median drift in the upper quatile (of what?) exceed the total median drift length? was there always wind shear present? I couldn't quite follow this sentence.
lline 23: were -> where
line 24: replace "and be able to release sondes" with ", where the sondes could be released."
line 28: include 'lower tropospheric' before 'profile'

p.7
lines 2-3 can be removed, this has already been stated.
over all the data description is okay, but the quicklooks, especially of the Tb, do not provide much information. An example of, e.g., the ability to retrieve a temperature profile, with a comparison to a dropsonde, would be nice.

p.8 line 7:  add "of clear skies" after "profiles"
line 25: rewrite sentence as "The erroneous time stamps were reconstructed where possible"
line 29-30: rewrite the 2nd phrase to remove the passive voice. 'appeared to be' -> are. remove 'about'
line 31: add 'angles' after 'bank'

p9
line 1: 'They' -> The comparisonl
ine 22: add 'the' before 'height'

p. 10
line 9: corresponding -> correspond

p. 11 line 9: has-> have

Table 1: 'shallow trade wind convection' occurs within every Flight charateristcs.  consider putting this in the table caption and restricting the characteristics to more differentiating features.

Table 5: "BAHAMAS" should be defined. not everyone will read the main text.

---

## Referee Comment (RC2) · John Bates (Referee) · 13 Mar 2019

The authors are to be commended for making this field campaign data set available together with extensive quality control and calibration data. Getting everything to work in a field campaign is always a combination of hard work and luck. Things happen and those involved must respond and be innovative to address calibration, navigation, and a myriad of logistical concerns. This article describes in detail, yet concisely, how all these concerns and challenges are this is done for the HAMP instruments. Significance The combination of passive microwave/millimeter wave instruments together with a cloud radar at 35 GHz combined with dropsonde data are powerful and unique

data sets. Passive microwave/millimeter wave instruments similar to HAMP have flown on both research and operational satellites now for almost 2 decades. A major challenge for the climate modeling and numerical weather prediction communities remains the parameterization of cloud processes and, particularly for NWP, the assimilation of cloudy radiances. This data set is useful for both communities due to the combination of active, passive, and in situ sounding data. The data set contains all elements needed by external users. Data are in the listed DOI repository and NetCDF formal and names/units are used. The data repository contains essentially an NCDUMP of the file headers which helps users ensure they know the file format in advance. The data are binned into common steps so that external users do not have to collocate the different data. Overflights by the NASA A-train, which include similar instruments, are identified. There is an opportunity to further enhance the utility of the data for the NWP community. Although publishing this data set is important, I would urge the authors to consider additional outreach. For NWP, they might contact the NWP satellite applications facility (https://www.nwpsaf.eu/site/) advertising this data set for 1-dimentional radiance assimilation testing with their microwave imaging processing package (https://www.nwpsaf.eu/site/software/mwipp/). For additional outreach to the climate community, the authors may try contacting the Copernicus Climate Change Service hosted by ECMWF (https://climate.copernicus.eu/) and the CMIP group in the USA at PCMDI (https://pcmdi.llnl.gov/index.html) as well as the Earth System Grid Federation https://esgf.llnl.gov/ ). ( I know that co-author Stevens has been active in this area as has Prof. Bony who appears to be a lead organizer of the Field campaign. Data Quality Since I am semi-retired, I do not have access to the full suite of tools the review criteria call for to assess completely the identified criteria. That said, a quick look at the data shows the fields to be within reasonable quality limits and the extensive discussion suggests the data are of high quality. It would be helpful for the authors to identify a potential physical mechanism for the application of the global bias identified in section 5.1 between the forward radiative transfer Tb computed from the profile data and the microwave measurements. Perhaps this is still under study. Microwave observations are sometimes subject to side lobe contamination biases that can be very difficult to identify and correct. Perhaps some others can comment on exercising the data. Presentation quality The manuscript has a nice balance of description required to use the data without going too long. Figure 1 and Table 1 are particularly helpful since they quickly help potential users identify different weather/climate regimes. The references of Bony et al. and Stevens et al. provide the more complete context for the field campaigns and scientific questions being studied. The 'find data' button on the doi landing page does take the user to the WDC Climate page for each NetCDF data set for each of the four campaigns. That page does contain an NCDUMP of the file header information that is helpful. Although there are no other quick look tools provided, I know from my experience that NetCDF tools are extensive and widely supported, so this is not a major issue.

---

## Author Comment (AC1) · 28 May 2019

**Authors' Response**

We would like to thank both reviewers for their helpful comments on the manuscript. The detailed questions and suggestions helped improving the manuscript a lot.

In the following, we will answer the comments of both reviewer in one document, starting with reviewer 1. Italics indicate reviewer comments and changes to the manuscript are in blue font. Page numbers and line number reference the originally submitted manuscript.

Following the specific answers, we've attached the updated manuscript with the changes marked in color. Red indicates parts that have been removed and blue indicates parts that have been added.

Comments from Reviewer #1

**My main 3 comments are that**

1) it would be nice to see an example of at least one scientific application of the dataset, including retrieved liquid water path, water vapor paths, radar/lidar derived cloud properties. Figures 2 and 3 are good as it is also nice to see what the actual quicklooks look like. These also communicate that the retrieval work remains to be done. Are there any examples of retrieved physical quantities from any of the flights that could be mentioned? Or any scientific results, perhaps already published in other papers? This could be considered to fall outside the scope of the ESSD journal, and if the editor also feels that way, then so be it, but it would help increase the impact of the manuscript to show the reader why they might be interested in working with it.

Since the scope of the ESSD journal are articles on original research data sets. We focus in this manuscript only on the level 1 data that is quality controlled and published along with the manuscript. However, retrieval development with this data has been already started and we agree that an example of the possible products might be very helpful for the user. We therefore included an additional figure (Fig. 3 in the updated manuscript) from the publication about the developed retrievals for liquid water path (Jacob, et al., 2019) to demonstrate the potential of the data. We extended the text in the manuscript to include this:

In summary, the combination of measurements from different channels can be used to derive the integrated water vapor and liquid water path (Schnitt et al., 2017; Jacob et al., 2019), coarse resolution temperature and moisture profiles as well as information on ice and snow occurrence. An example of retrieved liquid water path and rain water path is shown in Fig. 3 to demonstrate the potential of the HAMP measurements. This scene from the NARVAL1 campaign shows the time series of the retrieved quantities together with radar reflectivity measurements. The retrieval algorithm is described in depth in Jacob et al. (2019).

2) My second comment is that the manuscript contains many small grammatical inconsistencies. These merely reflect that the authors are not native English speakers and the meaning is generally clear, but if a native English speaker could be found to read through the manuscript and improve the English, that would be worth doing.

We thank the reviewer for their comments and suggestions about the wording of the manuscript. We asked a native speaker to review the manuscript to improve the language and applied their comments to our manuscript.

3) My third comment is that the description of the Vaisala dropsondes, "RD94" on p. 6, doesn't mean anything to mean. Some description and a reference to their performance would be nice. Is a

correction incorporated for a solar radiation bias, as is done for research grade radiosondes in the US? Are there any issues for the user to be aware of?

We have extended the description of the dropsondes in the manuscript by values of measurement accuracies and the relevant citations.

**The extended section about the dropsondes now reads:**

Vaisala RD94 dropsondes (Busen, 2012) were deployed in all campaigns using the Airborne Vertical Atmospheric Profiling System (AVAPS, Hock and Franklin, 1999). Wang et al. (2015) and Vaisala (2017) report that the measurement accuracy of these sondes for pressure is 0.4 hPa, temperature is 0.2 °C and relative humidity is 2%. The accuracy of horizontal wind speed measurements is estimated to be 0.1 m s-1.

We are not aware of any applications of corrections for solar radiation bias. We would however argue that this effect would be almost negligible in the case of this data set: The dropsondes were mainly dropped from around 8 km altitude. Average descent rate of dropsondes through the troposphere is about 10 m/s while average ascent rate of radiosondes in the troposphere is around 5 m/s. The time the dropsondes are influenced by solar radiation is therefore much shorter than for radiosondes.

According to the solar radiation correction table for RS92 radiosondes on the Vaisala webpage (https://my.vaisala.net/en/meteorology/products/soundingsystemsandradiosondes/soundingdatacontinu ity/RS92-Data-Continuity/Pages/solarradiationcorrectiontable.aspx, accessed on 23. May 2019), the correction values for the lower troposphere, where the majority of the dropsondes have been dropped, are rather small: between 0.04 °C for low solar angles and 0.14 °C for higher angles. This is below the reported accuracy of the temperature sensor. In our opinion, this, together with the faster descends of the dropsondes, this justifies not correcting the temperature measurements for solar radiation biases.

**Minor comments:**

page 1, line 21: "the full atmospheric column...." this depends on the frequency, would be worth stating.

**We have added this to the text the sentence now reads:**

Microwave frequencies are especially suited for cloud and precipitation remote sensing as they can, depending on the individual frequencies, penetrate the full atmospheric column in contrast to solar and infrared remote sensing which are mainly limited to thin clouds and cloud top regions.

page 2, line 4-5: "12 dBZ....typical for light precip.." is confusing. How about : "12 dBZ, more suited for the detection of heavier precipitation, and less so for characterizing shallow convection."

**Thank you for this suggestion. We have changed the sentence to:**

Passive microwave satellite instruments have footprints of several tens of kilometers (Elsaesser et al., 2017) while active microwave instruments are confined to narrow scan regions, limited vertical resolution and sensitivity, i.e. the Global Precipitation Mission (GPM) 35 GHz radar has a minimum detectable signal of 12 dBZ (Skofronick-Jackson et al., 2013), more suited for the detection of heavier precipitation, and less so for characterizing shallow convection.

p. 2, line 6: to -> with p. 2 line 22: put "The" before "Center" line 24: put 'was' before 'again'

We have corrected the points mentioned above.

lines 27,28: the verb tenses are not consistent.

We have changed the sentence to:

HAMP observations from NAWDEX give insight into convective clouds and their surroundings in the frontal regions and warm sector of mid-latitude cyclones.

p. 3, line 16: "collocating tracks": the word 'collocating' is used a lot within the manuscript. This is the first instance. At first I was not sure what the authors meant, it is not a typical usage. Perhaps rephrase as "and to evaluate satellite observations through flying underneath the satellite, along the projected satellite track near the satellite overpass time". Once this is clearly defined the continued use of 'collocated' will be clear I think.

We thank the reviewer for pointing this out. We changed the first mention of collocation to the following sentences:

Additionally, the focus of this campaign was to assess the representativeness of ground-based measurements on Barbados (Stevens et al., 2016) for a broader trade wind region over the tropical Atlantic and to evaluate satellite observations on collocated tracks (Klepp et al., 2014; Stevens et al., 2016). This has been done by flying underneath the satellite, along the projected satellite track near the satellite overpass time.

line 28: were -> was
p. 4, line 5: in -> for a
line 6: always precede "Center" with a "The".
p.5, line 10: with -> at, in -> while

We have corrected the points mentioned above.

*p.6 line 20: How can the median drift in the upper quartile (of what?) exceed the total median drift length? was there always wind shear present? I couldn't quite follow this sentence.*

We thank the reviewer for pointing out, that this sentence was hard to follow. The statement should be about the distribution of horizontal drift distances from all dropsonde profiles. The median of all horizontal drift distances is 3.8 km, the lower quartile of horizontal drift distances is 2.3 km and the upper quartile of these is 10.8 km. We have rewritten the sentence to the following:

The length of the horizontal drift distance varied substantially from drop to drop. The median of the horizontal drift distances for all dropsonde measurements was 3.8 km, while the upper and lower quartile of horizontal drift distances was 10.8 km and 2.3 km, respectively.

*line 23: were -> where*

*line 24: replace "and be able to release sondes" with ", where the sondes could be released." line 28: include 'lower tropospheric' before 'profile' p.7 lines 2-3 can be removed, this has already been stated.*

We have corrected the points mentioned above.

over all the data description is okay, but the quicklooks, especially of the Tb, do not provide much information. An example of, e.g., the ability to retrieve a temperature profile, with a comparison to a dropsonde, would be nice.

We completely agree, that quicklooks from brightness temperatures are hard to interpret. As we mentioned in the beginning, we focus this manuscript on the description of the original data set and their quality control procedures. To give a better insight into the potential of the data, we added Figure 3 to the manuscript which shows retrieved liquid water path and rain water path in comparison with radar reflectivity measurements from Jacob et al. (2019).

A lot of the evaluation of the data and retrieval development for profiles is still ongoing work. Unfortunately, we can not give an example for this as this has not been finished yet. But we would still like to publish the data set and make it available for the community, to give others the opportunity to use this data set for developing and testing their own retrievals. p.8 line 7: add "of clear skies" after "profiles"

*line 25: rewrite sentence as "The erroneous time stamps were reconstructed where possible" line 29-30: rewrite the 2nd phrase to remove the passive voice. 'appeared to be' -> are. remove 'about' line 31: add 'angles' after 'bank'*

We have corrected the points mentioned above.

*p9 line 1: 'They' -> The comparison*

We have rewritten the sentence: Ewald et al. (2018) concluded that the resulting bias of 7.6 dBZ originated from differences in software configuration and instrument calibration.

line 22: add 'the' before 'height' p. 10 line 9: corresponding -> correspond

We have corrected the points mentioned above.

**p. 11 line 9: has-> have**

We did not find the word "has" on page 11, line 9. We did, however, find that the word "has" in line 14 on the same page should be changed to "have" to be consistent in the use of the word "data" as plural throughout the manuscript. We changed an additional instance of this on page 9, line 9.

*Table 1: 'shallow trade wind convection' occurs within every Flight characteristics. consider putting this in the table caption and restricting the characteristics to more differentiating features.*

We thank the reviewer for this suggestion. The table caption now reads: List of NARVAL-South flights. All flights include observations of shallow trad wind convection. Additional flight characteristics are listed in the last column.

Table 5: "BAHAMAS" should be defined. not everyone will read the main text.

We added the explanation of the synonym (BAHAMAS: BAsic HALO Measurement And Sensor system) as a footnote to the table.

**Comments from Reviewer #2**

Although publishing this data set is important, I would urge the authors to consider additional outreach. For NWP, they might contact the NWP satellite applications facility (https://www.nwpsaf.eu/site/) advertising this data set for 1-dimensional radiance assimilation testing with their microwave imaging processing package (https://www.nwpsaf.eu/site/software/mwipp/). For additional outreach to the climate community, the authors may try contacting the Copernicus Climate Change Service hosted by ECMWF (https://climate.copernicus.eu/) and the CMIP group in the USA at PCMDI (https://pcmdi.llnl.gov/index.html) as well as the Earth System Grid Fed- eration https://esgf.llnl.gov/ ). ( I know that co-author Stevens has been active in this area as has Prof. Bony who appears to be a lead organiser of the Field campaign.

We thank the reviewer for pointing out the possible applications in the NWP and climate community. This is in part the reason why we decided to publish the data via the CERA database. The database assures good findability of the datasets by requiring detailed metadata for all published dataset. These metadata are then indexed by different other databases for a broader reach across the communities. For example, the CERA database is part of the Global Information System Centres (GISC) via the German Weather service. The data is also findable via the EUDAT collaborative data infrastructure thanks to the standard of the CERA database. The published HAMP dataset is therefore already accessible quite well. We will however keep this in mind and check if this setup is sufficient or if other steps might be necessary to advertise the data set to an even broader community.

It would be helpful for the authors to identify a potential physical mechanism for the application of the global bias identified in section 5.1 between the forward radiative transfer Tb computed from the profile data and the microwave measurements. Perhaps this is still under study. Microwave observations are sometimes subject to side lobe contamination biases that can be very difficult to identify and correct. Perhaps some others can comment on exercising the data.

Unfortunately we don't have a definitive answer to this. We are aware of the fact, that our measurements are not perfect. Microwave radiometer instruments on an aircraft are unfortunately always subject to vibration and temperature changes which can influence the behavior of the instruments in an unpredictable way.

We don't think that contamination from side lobes would be a large problem, since the environment in which the measurements are conducted is usually very homogenous. And so, even if there were influences from the sides, this should be not too different from the measurements directly nadir below the aircraft.

Up until now, we can't ultimately explain this bias. We found biases in the data systematically for all flights. But even after all our analyses, uncertainties still remain and the exact reason for these differences are still unclear. We decided therefore to thoroughly identify the biases in our measurements, report on all our findings, and include the information in the data files. Our main result from this investigation is, that a calibration against at least one clear-sky dropsonde per flight is needed.

1Universität Hamburg, Hamburg, Germany

[revised manuscript text omitted]